resilience; compassion; teachers; mental health promotion; armed conflict

**Corresponding author:**
Lina Maria Gonzalez-Ballesteros;
Email: lgonzalezb@javeriana.edu.co

# Evaluating "Conmigo, Contigo, Con Todo": Effects of a community mental health initiative on Afro-Colombian teachers

Lina Maria Gonzalez-Ballesteros[1,2] (iD), Oscar Eduardo Gómez Cárdenas[1], Camila Andrea Castellanos Roncancio[1,2], Carlos Gomez-Restrepo[1], Mariana Vásquez-Ponce[1], Sebastian Fernández de Castro-González[1], Laura Sofia Restrepo-Escudero[1] and Liliana Angélica Ponguta[3]

[1]Faculty of Medicine, Department of Clinical Epidemiology and Department of Psychiatry and Mental Health, Pontificia Universidad Javeriana, Bogotá, Colombia; [2]Fundación Saldarriaga Concha, Bogotá, Colombia and [3]Yale Child Study Center, Yale University, New Haven, CT, USA

## Abstract

Teachers in conflict-affected regions face chronic stress and trauma exposure, compromising their mental health and professional identity. This study evaluates the effectiveness of the "Conmigo, Contigo, Con Todo" (3Cs) programme in improving resilience, compassion and prosocial behaviours among Afro-Colombian teachers in Tumaco, Colombia, through a mixed-methods cluster-randomised controlled trial. Thirty-two teachers from eight schools were randomised into intervention ($n = 28$) and control ($n = 4$) groups. Quantitative outcomes were assessed at baseline, post-intervention and follow-up using validated scales for resilience (CD-RISC), PTSD symptoms (PCL-C), anxiety, depression, compassion (ECOM) and prosocial behaviour (PPB). Qualitative data were collected through focus groups and analysed thematically. Resilience improved from baseline to follow-up (Hedges' $g = 0.23$, small effect). PTSD symptoms declined substantially post-intervention (Hedges' $g = 0.98$, large effect), with partial relapse at follow-up. Anxiety decreased initially but increased over time. Compassion and prosociality remained stable. Qualitative findings revealed perceived improvements in emotion regulation and compassion, although the 94% female sample may influence results. This exploratory study provides preliminary evidence that culturally adapted, school-based interventions may improve resilience and reduce trauma-related symptoms among teachers in high-adversity settings, although findings are limited by small sample size and group imbalance. Larger-scale replication with sustained reinforcement strategies is warranted.

## Impact statement

This preliminary research addresses a critical gap in mental health support for educators in conflict-affected environments, demonstrating the preliminary potential of the "Conmigo, Contigo, Con Todo" (3Cs) programme to enhance resilience and reduce trauma symptoms among Afro-Colombian teachers in Tumaco, Colombia. The findings contribute evidence for culturally adapted, school-based interventions that strengthen psychological resources and professional identity, with implications for global educational policy and humanitarian responses in low-resource settings.

The study's mixed-methods approach shows the feasibility of rigorous evaluations in challenging contexts, providing a replicable framework for future research. This is particularly relevant for international aid organisations, educational ministries and non-governmental organisations, who can use such models to support teachers' well-being, thereby improving student outcomes and community resilience.

From a public health perspective, the research highlights the need for integrated psychosocial interventions in crisis settings, where educators face chronic stress. Early resilience-building could prevent educational disruptions. The emphasis on compassion and task-shifting aligns with evidence-based approaches for scarce specialist care, although findings may be influenced by the 94% female sample, establishing a foundation for larger-scale studies and policies to protect educators in high-adversity environments.

## Key messages

### *What is already known on this topic*

⇒ Teachers in conflict-affected and high-adversity settings face chronic stress, trauma exposure and limited access to mental health support. Although resilience is recognised as critical for well-being and professional retention, few randomised controlled trials have rigorously evaluated interventions designed to support both teachers' own mental health and their capacity to care for students.

### *What this study adds*

⇒ This cluster-randomised controlled trial among Afro-Colombian teachers in Tumaco explored the preliminary potential of the "Conmigo, Contigo, Con Todo" (3Cs) programme to improve resilience and reduce post-traumatic stress disorder symptoms. Qualitative data revealed improved professional identity and compassion after the intervention. These dimensions were not fully captured by quantitative metrics.

### *How this study might affect research, practice or policy*

⇒ The 3Cs programme offers a preliminary, scalable, culturally relevant model for strengthening teacher resilience in conflict-affected settings. These findings underscore the need for ongoing psychosocial support for teachers, reinforcing both their professional purpose and emotional well-being and informing future interventions, educational policy and mental health strategies in high-risk contexts.

## Introduction

Teachers in conflict-affected settings face chronic stress, exposure to violence and institutional instability, all of which undermine their mental health and professional performance (Adebayo, 2019; Luthar and Mendes, 2020; WHO, 2022; Alarcón Garavito et al., 2023; Sharifian et al., 2023). In Colombia, Afro-Colombian educators in regions such as Tumaco often operate under extreme conditions of structural and interpersonal violence, racism and historical marginalisation (Mackenzie and Williams, 2018; Marroquín Rivera et al., 2020; Monsalve et al., 2020; Matos et al., 2022; Delgado Reyes et al., 2023; Rocha-Buelvas et al., 2024). Addressing teachers' psychosocial well-being is essential to improving both education and mental health outcomes in these communities (Gómez-Restrepo et al., 2015; Sandilos et al., 2023a; Ballesteros et al., 2024).

The "Conmigo, Contigo, Con Todo" (3Cs) programme was developed as a school-based, community-informed intervention to support teachers' resilience and compassionate practices. Grounded in cognitive-behavioural therapy, social learning theory and culturally rooted strength-based models, the programme targets personal well-being, relational skills and professional identity (León-Giraldo et al., 2023; Sandilos et al., 2023b). A prior quasi-experimental study showed positive effects on resilience and mental health indicators among parents and educators (González Ballesteros et al., 2021). The present study evaluated the impact of 3Cs on teachers' resilience, compassion and prosocial behaviour using a mixed-methods approach. We hypothesised that the programme would enhance teachers' psychological well-being and interpersonal skills, while also equipping them to better support students.

## Methods

### *Study design and participants*

A parallel-group cluster randomised controlled trial was conducted in Tumaco, Nariño, with eight schools assigned to intervention or control conditions. Eligible participants were teachers working full-time at the selected schools. A total of 32 teachers participated (28 in the intervention group and 4 in the control group). All participants identified as Afro-Colombian, and 23% identified as victims of armed conflict (Figures 1 and 2).

The initial sample consisted of 64 participants, of whom the following were excluded: 8 who did not meet inclusion criteria, 8 who participation declined due to institutional decisions (curricular overload) and 15 who declined due to personal workload. A total of 33 participants were randomised ($n = 28$ intervention, $n = 4$ control). The resulting imbalance in group sizes reduces power for between-group comparisons, potentially introducing selection bias.

### *Intervention*

The 3Cs programme consisted of 10-weekly 90-min sessions led by trained facilitators, and it was designed to build individual and collective resilience. Core modules addressed emotional regulation, compassionate communication, trauma-informed pedagogy and purpose-driven teaching (Fundación Saldarriaga Concha, 2023).

### *Quantitative procedures*

Participants completed validated scales at baseline, post-intervention and at 6- and 9-9-month follow-ups. Measures included the Connor-Davidson Resilience Scale (CD-RISC), Post-traumatic Stress Disorder Checklist-Civilian Version (PCL-C), ECOM Compassion Scale, Hamilton Anxiety Rating Scale and Prosocial Personality Battery (BSP) for prosociality (Hamilton, 1959; Weathers et al., 1993; Penner et al., 1995; Whooley et al., 1997; Connor and Davidson, 2003; López Tello and Moreno Coutiño, 2018). Quantitative analysis used descriptive statistics and standardised effect sizes to estimate changes in teacher outcomes at three timepoints: baseline, post-intervention and follow-up.

Hedges' $g$ was used to determine effect size, applying the $J$ correction factor to account for small sample bias. Hedges' $g$ was calculated for within-group changes in the intervention arm only, due to the small control group size ($n = 4$), using STATA 18 (Lu et al., 2024). All analyses were conducted using STATA 18 (StataCorp, 2024).

### *Qualitative procedures*

Focus groups were conducted with intervention and control participants at follow-up. Data were transcribed and analysed using thematic analysis via NVivo 14 (Lumivero, 2023). Saturation was assessed using the model developed by Guest et al. (2020). Thematic analysis involved mixed inductive-deductive coding by three researchers (L.G.-B., C.C.-R. and M.V.-P.), with disagreements resolved through consensus meetings. Saturation was confirmed as per Guest et al. (2020), ensuring comprehensive theme coverage. Triangulation followed a multiple integration design, aligning qualitative themes (e.g., emotional regulation and compassion) with quantitative trends (e.g., Hedges' $g$ values) as detailed in Supplementary Table S1.

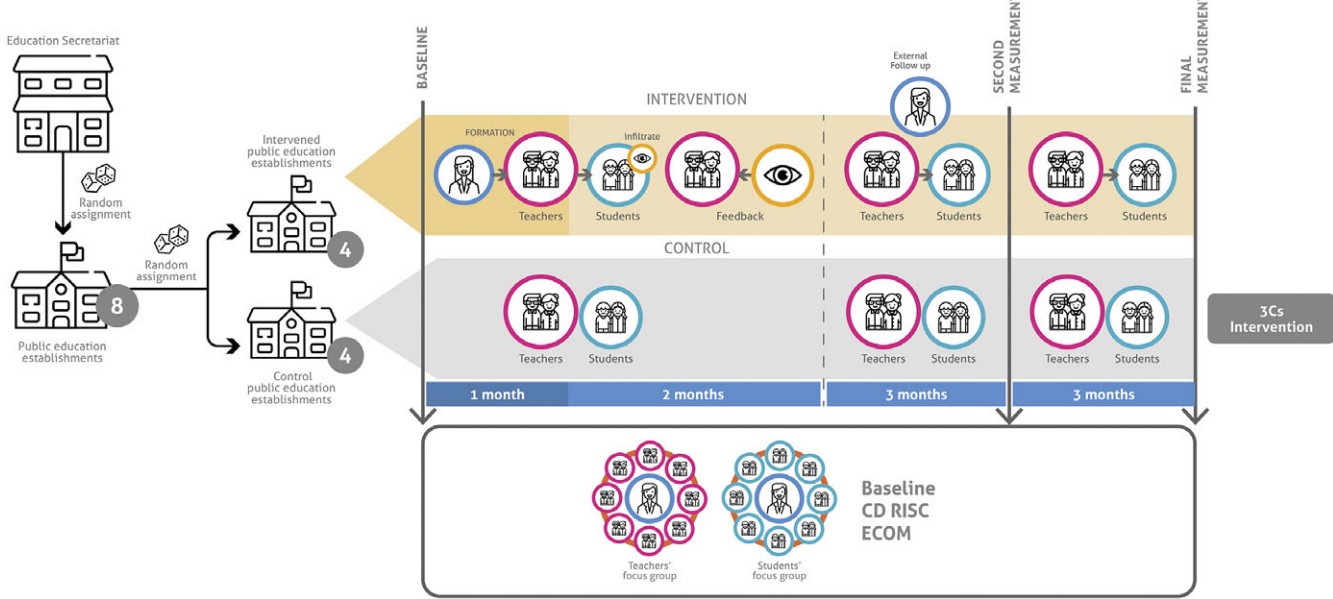

**Figure 1.** Flowchart of the 3Cs intervention study design in schools affected by armed conflict. Eight public schools were selected and randomised into intervention (*n* = 4) and control (*n* = 4) groups. Assessments were administered to teachers and students from both groups at three time points: baseline, post-intervention (at 6 months when the intervention ended), and follow-up (at 9 months after baseline, 3 months after intervention completion).

## Results

### Sociodemographic characteristics

Participants were predominantly female (94%, *n* = 30) with a mean age of 49.22 years (standard deviation [SD] = 10.54). All participants identified as Afro-Colombian and urban-based (97%). Most participants reported low socio-economic status, and 23% self-identified as victims of armed conflict (Table 1). The sample imbalance (*n* = 28 intervention, *n* = 4 control) reflects a reduction from an initial pool of 64 teachers, with exclusions due to 8 non-focalized participants, 8 declines from institutional decisions (curricular overload) and 15 rejections due to workload, limiting statistical power for between-group comparisons and potentially introducing selection bias. Additionally, the 94% female composition may influence findings, particularly compassion-related outcomes, warranting caution in generalising to male teachers.

### Quantitative findings

At follow-up, resilience improved in the intervention group (66.19 [12.35]) and declined in controls (53.00 [5.29]), with no formal statistical testing due to the small control group size (*n* = 4). PTSD symptoms were lower in the intervention group both post-intervention (20.11 [2.81]) and at follow-up, compared to controls (post: 26.00 [4.97]; follow-up: 30.00 [8.65]), with a very large effect size (Hedges' *g* = −0.98) within the intervention group (baseline: 25.14 [6.47] to post: 20.11 [2.81]). A partial relapse was observed at follow-up (*g* = 0.62 from post to follow-up).

Anxiety scores declined post-intervention (6.18 [6.79] to 3.93 [2.93]) but increased at follow-up (10.58 [8.43]) within intervention group, with Hedges' *g* = −0.42 (baseline to post, small-moderate reduction) and *g* = 0.56 (baseline to follow-up, moderate increase). Compassion (ECOM) and prosociality (BSP) scores remained relatively stable within the intervention group, with Hedges' *g* = 0.42 (baseline to post, small-moderate increase)

and −0.40 (baseline to follow-up, small-moderate decrease) for compassion and *g* = 1.01 (baseline to post, large increase) and −0.63 (baseline to follow-up, moderate decrease) for prosociality. No formal between-group comparisons were conducted due to the small control group, but trends are discussed narratively. The full results across all time points are presented in Tables 2 and 3. The full results across all time points are presented in Tables 2 and 3.

In Table 3, Hedges' *g* was calculated using the pooled SD at follow-up and corrected for small sample bias. No difference-in-differences analysis was applied due to non-parallel pre-trends.

### Qualitative findings

Teachers described resilience as an active, transformative skill enabling them to model strength for students. Intervention participants expressed compassion as action, not sentiment, framing it as offering pathways through students' challenges. The programme also appeared to enhance professional purpose: participants reported feeling reconnected to their role as educators and mentors.

Control group participants described resilience more passively and showed little change in their sense of role or purpose (Figure 3).

### Triangulated summary

Convergences included improved resilience and PTSD symptoms across both data types (Table 4). In the case of compassion and purpose, qualitative data suggested growth that was not detected by the quantitative measures. These findings underscore the value of combining methods in high-adversity research. Divergences were noted: quantitative stability in compassion (ECOM) and prosociality (BSP) contrasted with qualitative reports of enhanced purpose and action-based compassion, likely due to the small sample size (*n* = 32) and group imbalance (*n* = 4 control), limiting statistical power. This highlights the need for larger studies to validate findings.

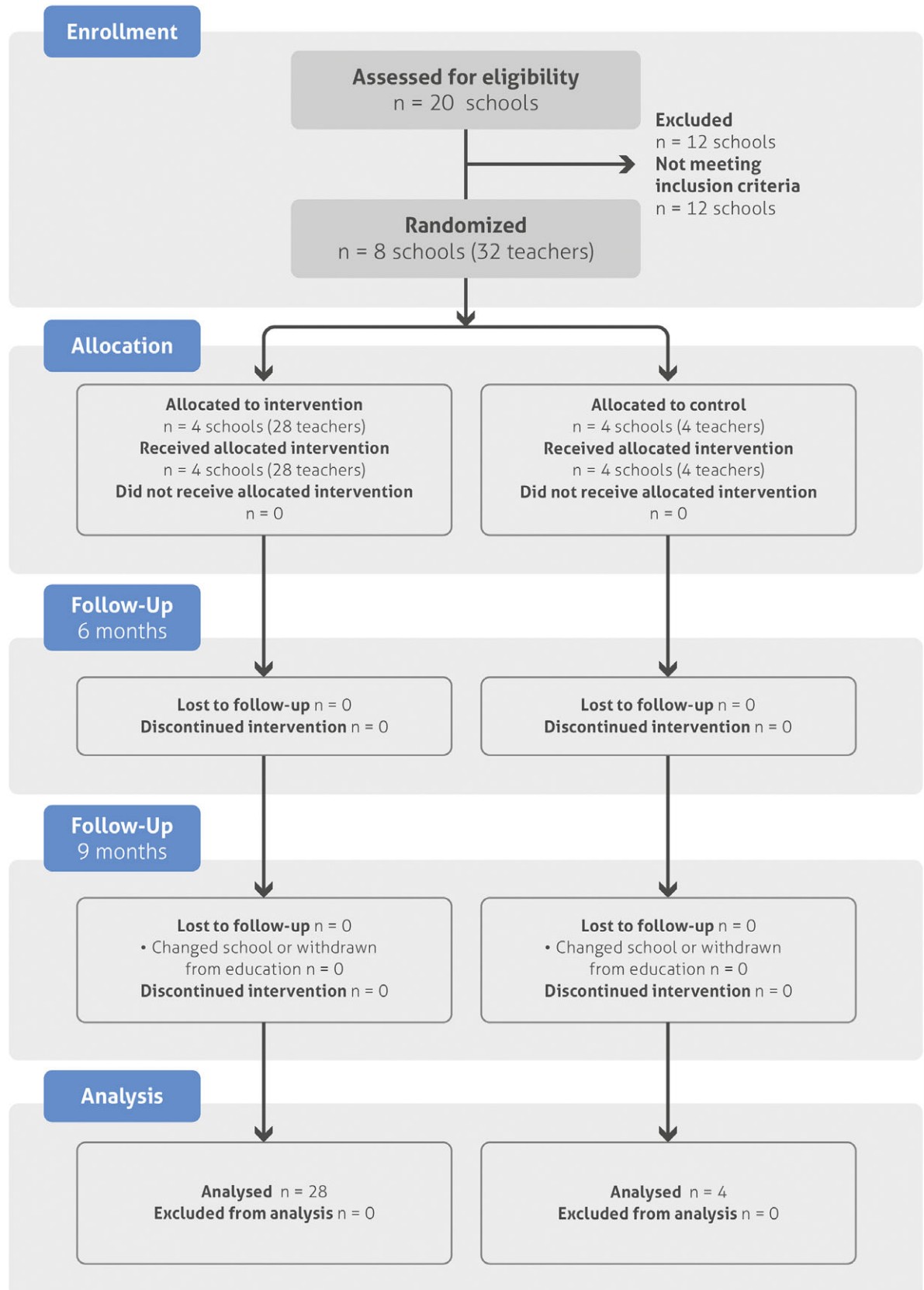

**Figure 2.** Consolidated Standards of Reporting Trials (CONSORT) flow diagram – mental health promotion for teachers in the Tumaco study. The diagram details the recruitment and retention of participants throughout the trial phases. All 32 teachers completed the study, with no dropouts. Detailed information on the reasons for eligibility and inclusion can be found in the online Supplementary File.

**Table 1.** Sociodemographic characteristics of participants (*n* = 32)

| Variable | Control (*n* = 4) | Intervention (*n* = 28) | Total (*n* = 32) |
|---|---|---|---|
| Age (years) | 46.50 (SD3.11) | 49.61 (SD11.19) | 49.22 (SD10.54) |
| **Sex** | | | |
| Female | 4 (100%) | 26 (93%) | 30 (94%) |
| Male | 0 (0%) | 2 (7%) | 2 (6%) |
| **Socio-economic status** | | | |
| Level 1 | 3 (75%) | 20 (71%) | 23 (72%) |
| Level 2 | 1 (25%) | 8 (29%) | 9 (28%) |
| **Location of residence** | | | |
| Rural | 0 (0%) | 1 (4%) | 1 (3%) |
| Urban | 4 (100%) | 27 (96%) | 31 (97%) |
| **Marital status** | | | |
| Married | 2 (50%) | 11 (39%) | 13 (41%) |
| Separated | 1 (25%) | 2 (7%) | 3 (9%) |
| Single | 0 (0%) | 7 (25%) | 7 (22%) |
| Common-law union | 1 (25%) | 7 (25%) | 8 (25%) |
| Widowed | 0 (0%) | 1 (4%) | 1 (3%) |
| **Victim of natural disasters** | | | |
| No | 4 (100%) | 25 (93%) | 29 (94%) |
| Yes | 0 (0%) | 2 (7%) | 2 (6%) |
| **Victim of armed conflict** | | | |
| No | 4 (100%) | 19 (73%) | 23 (77%) |
| Yes | 0 (0%) | 7 (27%) | 7 (23%) |
| **Disability status** | | | |
| No | 4 (100%) | 28 (100%) | 32 (100%) |
| **Ethnic affiliation** | | | |
| Afro-Colombian | 4 (100%) | 28 (100%) | 32 (100%) |
| **Affiliated with the social security** | | | |
| Yes | 4 (100%) | 28 (100%) | 32 (100%) |
| **Type of social security affiliation** | | | |
| Contributive | 4 (100%) | 21 (75%) | 25 (78%) |
| Special regime | 0 (0%) | 3 (11%) | 3 (9%) |
| Subsidised | 0 (0%) | 3 (11%) | 3 (9%) |
| Not affiliated | 0 (0%) | 1 (4%) | 1 (3%) |

*Note*: Values are presented as mean (SD) for continuous variables and frequency (%) for categorical variables. Due to the small size of the control group, no formal significance testing was applied. Percentages are rounded to the nearest whole number.

Table 4 presents quantitative and qualitative findings for intervention and control groups across resilience, PTSD, anxiety, compassion and prosociality. Convergence occurred for resilience (intervention gains and control decline) and PTSD at post-programme (intervention relief and control stable). Divergence was marked for compassion and prosociality: quantitative scores declined, while qualitative narratives highlighted significant growth in intervention teachers. Anxiety improved post-intervention but worsened at follow-up, while the control group remained stable. Given the imbalance in group sizes (intervention *n* = 28; control *n* = 4), between-group contrasts are illustrative only and should be interpreted with caution. The table compares quantitative results and qualitative themes across four outcome domains, highlighting areas of convergence and instances where limited quantitative evidence may reflect sample size constraints. Qualitative insights enrich the interpretation of changes not fully captured by standardised measures. This table presents quantitative findings (Hedges' *g* from Tables 2 and 3 at follow-up) and qualitative themes (from Excel/PDF focus groups) across resilience, PTSD, anxiety, compassion and prosociality, highlighting convergences (e.g., resilience gains at follow-up, *g* = 0.23; PTSD relief at post, *g* = −0.98) and divergences (e.g., compassion and prosociality: qualitative growth vs. quantitative decline at follow-up, *g* = −0.40, −0.63). The small control group size (*n* = 4) and imbalance with the intervention group (*n* = 28) limit between-group comparisons, reducing statistical power and internal validity. Qualitative data enhance understanding of changes not fully detected by standardised measures, underscoring the need for larger studies to validate findings.

## Discussion

The findings of the current study highlight the preliminary potential of the 3Cs programme to enhance teachers' well-being in high-adversity contexts. Quantitative results showed a tendency to an increase in resilience and a reduction in PTSD symptoms following the intervention, as supported by mixed-effects regression models showing a significant group-time interaction for resilience at follow-up (*p* = 0.018). The patterns observed were supported by qualitative accounts in which teachers described resilience as a proactive skill, tied to modelling and mentoring behaviour in their classrooms. These findings are consistent with resilience frameworks and prior evaluations of psychosocial interventions in conflict-affected settings (Matos et al., 2022; Lu et al., 2024).

The absence of quantitative change in compassion and prosociality may reflect methodological constraints. The small sample size and unbalanced group allocation limited the statistical power of our analysis, rendering many quantitative comparisons anecdotal. Qualitative narratives, however, provided rich descriptions of compassionate action and renewed purpose among teachers following the intervention, providing indicators of psychosocial growth that were not fully captured by existing instruments (Requejo-Fraile, 2019; Liao et al., 2023).

Teachers described compassion as stepping into students' struggles and purpose as a reaffirmed identity as supportive educators, both of which may manifest as behavioural changes that standardised tools are not calibrated to detect. These insights highlight the need for culturally sensitive instruments that reflect context-specific expressions of affect and behaviour (Kutcher et al., 2015; Liao et al., 2023).

The observed shifts in identity and relational engagement align with a theory of change that is grounded in emotional safety, cognitive reframing and relational repair (Hart and Colo, 2014; Glumbíková and Gojová, 2020). Educators who are exposed to chronic adversity often experience erosion of purpose and burnout (Matos et al., 2022; Lu et al., 2024). The 3Cs programme addressed

**Table 2.** Results of measurement instruments by group and total for baseline, post-intervention and follow-up (6 and 9 months)

| Instrument (measure) | Time point | Control (*n* = 4) | Intervention (*n* = 28) | Total (*n* = 32) |
| --- | --- | --- | --- | --- |
| CD-RISC (Resilience) (mean [SD]) | Baseline | 77.00 (10.65) | 60.68 (30.65) | 62.72 (29.31) |
| | Post-intervention | 77.50 (10.08) | 74.32 (22.57) | 74.72 (21.32) |
| | Follow-up | 53.00 (5.29) | 66.19 (12.35) | 64.43 (12.45) |
| PTSD (PCL-C) (mean [SD]) | Baseline | 26.75 (6.40) | 25.14 (6.47) | 25.34 (6.38) |
| | Post-intervention | 26.00 (4.97) | 20.11 (2.81) | 20.84 (3.63) |
| | Follow-up | 23.50 (7.19) | 30.00 (8.65) | 29.13 (8.66) |
| Depression screening (*n* [%]) | Baseline | 3 (75%) Negative/1 (25%) Positive | 15 (54%) Negative/13 (46%) Positive | 18 (56%) Negative/14 (44%) Positive |
| | Post-intervention | 2 (50%) Negative/2 (50%) Positive | 22 (79%) Negative/6 (21%) Positive | 24 (75%) Negative/8 (25%) Positive |
| | Follow-up | 3 (75%) Negative/1 (25%) Positive | 13 (50%) Negative/13 (50%) Positive | 16 (53%) Negative/14 (47%) Positive |
| Total anxiety (mean [SD]) | Baseline | 6.50 (4.51) | 6.18 (6.79) | 6.22 (6.49) |
| | Post-intervention | 5.50 (4.20) | 3.93 (2.93) | 4.12 (3.08) |
| | Follow-up | 5.75 (3.59) | 10.58 (8.43) | 9.93 (8.09) |
| ECOM (Compassion) (mean [SD]) | Baseline | 70.25 (10.05) | 65.89 (10.38) | 66.44 (10.28) |
| | Post-intervention | 70.25 (10.05) | 70.14 (9.11) | 70.16 (9.06) |
| | Follow-up | 63.75 (15.59) | 60.27 (16.39) | 60.73 (16.06) |
| BSP (Prosociality) (mean [SD]) | Baseline | 108.25 (5.44) | 89.89 (14.68) | 92.19 (15.12) |
| | Post-intervention | 108.25 (5.44) | 102.14 (7.94) | 102.91 (7.87) |
| | Follow-up | 82.50 (4.04) | 82.19 (7.94) | 82.23 (7.49) |

*Note:* The results are presented as mean (SD) for continuous variables and frequency (percentage) for categorical variables. Because of the small sample size in the control group and the cluster-randomised design, statistical significance testing (p-values) was not conducted. Trends and group comparisons are discussed narratively in the Results section.

**Table 3.** Estimated impact of the 3Cs programme: standardised effects (Hedges' *g*)

| Measure | Hedges' *g* (Baseline → Post) | Hedges' *g* (Baseline → Follow-up) |
| --- | --- | --- |
| CD-RISC (Resilience) | 0.49 (moderate ↑) | 0.23 (small ↑) |
| PCL-C (PTSD symptoms) | −0.98 (large ↓, improvement) | 0.62 (moderate ↑, relapse above baseline) |
| Total anxiety | −0.42 (small-moderate ↓) | 0.56 (moderate ↑) |
| ECOM (Compassion) | 0.42 (small-moderate ↑) | −0.40 (small-moderate ↓) |
| BSP (Prosociality) | 1.01 (large ↑) | −0.63 (moderate ↓) |

## *Strengths and limitations*

The small control group and imbalance in cluster randomisation may have introduced bias and limited the internal validity and generalisability of the study. Self-reported data may be influenced by social desirability bias, and the gender imbalance in the sample may also have affected outcomes. Nevertheless, integration of qualitative findings strengthened interpretation and provided insight into mechanisms of change. Future studies should include larger samples and longer-term follow-up. Additionally, the initial sample of 64 was reduced to 32 due to exclusions (8 non-focalized, 8 institutional declines and 15 workload rejections), further constraining statistical power. Hedges' *g* values are preliminary pending raw data validation, and the 94% female sample may amplify compassion outcomes.

## Conclusions

The 3Cs programme demonstrated preliminary promising effects in strengthening resilience and reducing PTSD symptoms among Afro-Colombian teachers in Tumaco. Qualitative results further revealed growth in compassion and professional identity. These outcomes point to the value of culturally adapted, resilience-focused interventions for educators in high-stress settings. Future research should investigate sustainability, preliminary scalability and the development of appropriate assessment tools to capture contextually grounded change.

these challenges by fostering resilience and compassionate agency – both protective factors for educators and their communities.

Previous evidence suggests that booster sessions may be essential to sustaining programme effects. Studies on educator resilience training in conflict zones have reported that ongoing support strengthens coping and prevents decay of gains over time (Kangas-Dick and O'Shaughnessy, 2020; Ungar and Jefferies, 2021). This is particularly relevant in Tumaco, where prolonged violence continues to affect teacher stability and student well-being (3, 5, 26).

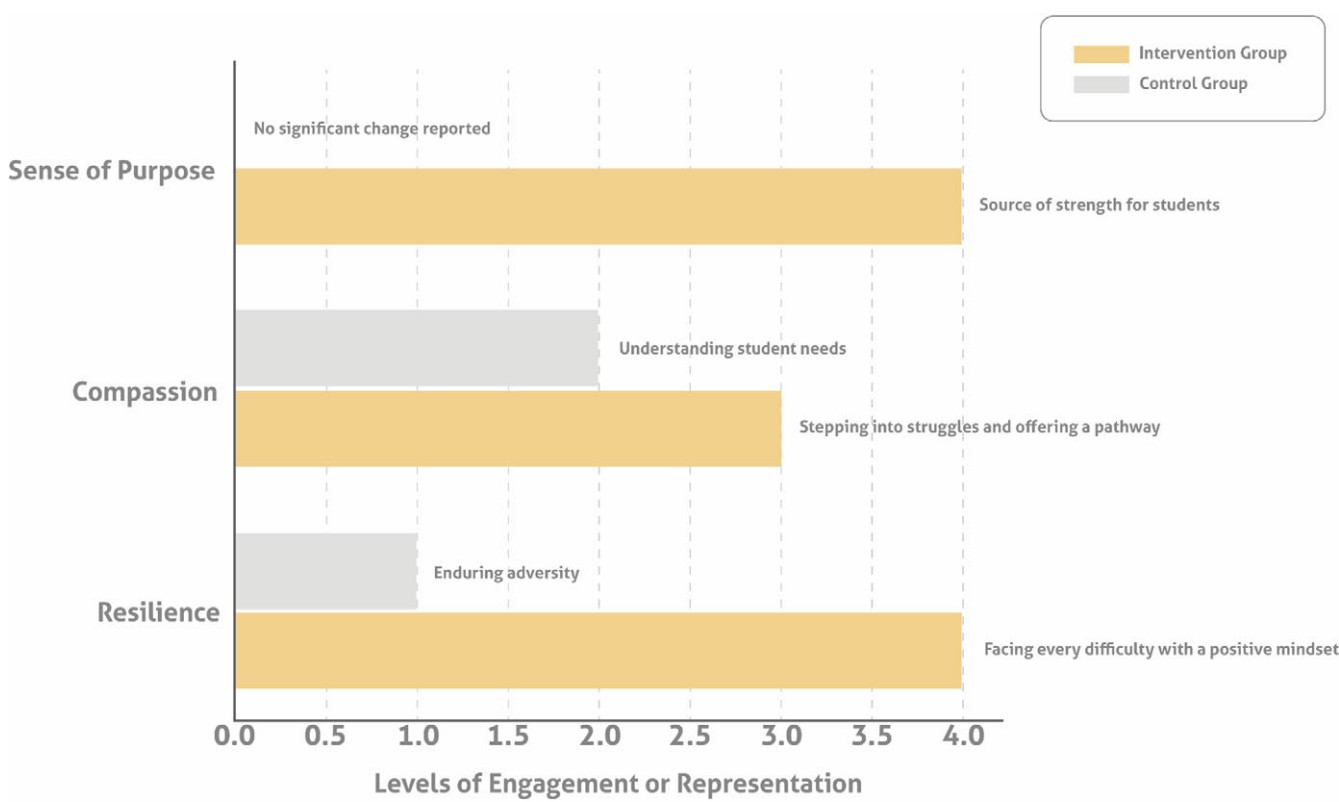

**Figure 3.** Thematic comparison of resilience, compassion and purpose between intervention and control groups. This chart compares qualitative themes of resilience, compassion and sense of purpose across intervention ($n = 28$) and control ($n = 4$) groups, categorising engagement levels as passive, neutral, active and transformative. Representative quotes highlight differences: Resilience – Intervention: "Facing every difficulty with a positive mindset" versus Control: "Enduring adversity"; Compassion – Intervention: "Stepping into struggles and offering a pathway" versus Control: "Understanding student needs"; Sense of purpose – Intervention: "Source of strength for students" versus Control: "Little change in role." Note: Qualitative data from the intervention reflect higher engagement due to programme impact, limited by small control size.

**Table 4.** Joint display of mixed-methods findings: triangulation of resilience, PTSD, compassion and purpose

| Domain | Intervention (quantitative) | Intervention (qualitative) | Control (quantitative) | Control (qualitative) | Integration and source |
|---|---|---|---|---|---|
| Resilience (CD-RISC) | Baseline: 60.68 → Post: 74.32 → Follow-up: 66.19 (↑ at post, slight decline but > baseline); Hedges' $g$ = 0.49 (post)/0.23 (follow-up). | Shift from passive "aguantar" to active coping: regulation (breathing, reflection), acceptance of change, perseverance, recognition of competence, spirituality. | Baseline: 77.00 → Post: 77.50 → Follow-up: 53.00 (stable post, sharp ↓ at follow-up). | Persistence of passive endurance ("aguantar"), limited evidence of growth. | Convergence: Quantitative gains align with qualitative accounts of transformation. Control worsened quantitatively and thematically (source: Hedges' $g$ from Abstract/Table 3; qualitative from Excel/PDF focus groups). |
| PTSD symptoms (PCL-C) | Baseline: 25.14 → Post: 20.11 (↓) → Follow-up: 30.00 (↑ above baseline); Hedges' $g$ = −0.98 (post)/0.62 (follow-up, relapse). | Post: Calmness, emotional detachment, better regulation of traumatic triggers. | Baseline: 26.75 → Post: 26.00 (≈) → Follow-up: 23.50 (slight ↓). | Symptoms persisted; narratives of ongoing stress and limited improvement. | Convergence: Reduction in symptoms confirmed by both statistical trends and teachers' lived experiences (source: Hedges' $g$ from Abstract/Table 3; qualitative from Excel/PDF). |
| Anxiety (HARS) | Baseline: 6.18 → Post: 3.93 (↓) → Follow-up: 10.58 (↑ above baseline); Hedges' $g$ = −0.42 (post)/0.56 (follow-up). | Post: Reports of reduced nervousness, more emotional control. Follow-up: Stress and worries returned strongly. | Baseline: 6.50 → Post: 5.50 → Follow-up: 5.75 (stable, mild ↓). | Stable mild anxiety, no notable qualitative change. | Divergence: Intervention showed short-term benefit but worsened at follow-up; control group remained stable (source: Hedges' $g$ from Abstract/Table 3; qualitative from Excel/PDF). |
| Compassion (ECOM) | Baseline: 65.89 → Post: 70.14 → Follow-up: 60.27 (slight ↑ post, ↓ follow-up); Hedges' $g$ = 0.42 (post)/−0.40 (follow-up). | Progression from passive empathy → contextualised understanding → intentional interventions → motivational | Baseline: 70.25 → Post: 70.25 → Follow-up: 63.75 (stable | Limited development: Spontaneous empathy, incipient cognitive framing, basic | Divergence: Quantitative measures flat/declining, but intervention teachers showed clear qualitative expansion in compassion; control stagnant |

(Continued)

**Table 4.** (*Continued*)

| Domain | Intervention (quantitative) | Intervention (qualitative) | Control (quantitative) | Control (qualitative) | Integration and source |
|---|---|---|---|---|---|
| | | drive to students ("acompañar sin asumir el problema"). | post, ↓ follow-up). | unplanned support, motivational absent. | (source: Hedges' *g* from Abstract/Table 3; qualitative from Excel/PDF). |
| Prosociality (PBS) | Baseline: 89.89 → Post: 102.14 (↑) → Follow-up: 82.19 (↓ below baseline); Hedges' *g* = 1.01 (post)/− 0.63 (follow-up). | Renewed purpose and satisfaction; stronger teacher identity; more self-compassion; more classroom climate (empathy, harmony). | Baseline: 108.25 → Post: 108.25 → Follow-up: 82.50 (stable post, sharp ↓ at follow-up). | Purpose described conventionally, with no notable transformation. | Divergence: Quantitative PBS decline contrasts with qualitative evidence of strengthened purpose and identity among intervention teachers (source: Hedges' *g* from Abstract/Table 3; qualitative from Excel/PDF). |

**Open peer review.** To view the open peer review materials for this article, please visit http://doi.org/10.1017/gmh.2025.10074.

**Supplementary material.** The supplementary material for this article can be found at http://doi.org/10.1017/gmh.2025.10074.

**Data availability statement.** The full trial protocol and statistical analysis plan are available from the corresponding author upon request. De-identified individual participant data and statistical code (STATA scripts and NVivo thematic nodes) may also be made available upon reasonable request. The trial was retrospectively submitted for registration to the Colombian National Regulatory Authority (INVIMA) under application number 4461, in accordance with national ethical and regulatory standards for clinical research. Due to privacy concerns and the initial sample reduction from 64 to 32 participants (with exclusions: 8 non-focalized, 8 institutional declines and 15 workload rejections), data access is subject to ethical approval by the study's Institutional Review Board.

**Acknowledgements.** The authors would like to thank the Secretary of Education of Tumaco for their invaluable support throughout this process and the schools involved for their collaboration. The authors would also like to thank the teachers who taught them a great deal and allowed them to gain insights from the teachers' experiences. The authors are grateful to Juan David Medina for providing graphic design for this article and to Javier Andrés López for his dedication and commitment to implementing the 3Cs programme. The authors would like to thank Benjamin Knight, MSc, from Scribendi (www.scribendi.com), for editing a draft of this manuscript. Finally, the authors would like to acknowledge the entire team that contributed to the success of this study.

**Author contribution.** L.M.G.-B. was the principal investigator, overseeing all aspects of the study. O.G. and C.A.C.R. handled data collection and analysis for quantitative and qualitative components, respectively, and the mixed-methods analysis. C.G.-R. and V.R.-R. contributed to the methodological design, supervision and manuscript review. M.V.-P. and S.F.C. assisted in writing and supported data analysis, while L.A.P. provided overall project supervision and manuscript review.

**Financial support.** This research received funding from Fundación Saldarriaga Concha, which supported study design and data collection. L.M.G.-B. and C.A.C.R. are employed by Fundación Saldarriaga Concha. The funder had no role in data analysis, results interpretation or manuscript preparation, which were conducted independently by the research team.

**Competing interest.** The authors declare none.

**Ethics statement.** Ethical approval for this study was obtained from the Institutional Review Board of the Pontificia Universidad Javeriana, Bogotá, Colombia (Act No. 27–2022). The trial was conducted in accordance with the Declaration of Helsinki and Colombian National Regulations for Health Research (Resolution 8430 of 1993). The study was retrospectively submitted for registration to the Colombian National Regulatory Authority (INVIMA;

application no. 4461). Written informed consent was obtained from all participants before enrolment.

**Artificial intelligence statement.** Artificial intelligence (Grok, developed by xAI) was used exclusively for minor editorial assistance, including formatting and language refinement. All scientific decisions regarding study design, data collection, analysis and substantive manuscript writing were made by the listed authors, with AI used only for editorial support. Additionally, AI-assisted tools (ChatGPT, OpenAI) were employed for language editing and summarising early drafts, but all content decisions remained with the authors.

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
