## [Reviewer Report]

This is an interesting and valuable study, especially in its effort to integrate mixed methods in work with vulnerable communities. Thank you for the opportunity to review it. My comments focus on clarifying the limitations created by the imbalance between the intervention and control groups, improving the explanation about qualitative analysis and triangulation process, and being more cautious with claims about scalability. I also suggest discussing the implications of the gender imbalance in the sample and making Table 4 clearer in terms of what is being integrated. Here are my comments:

Page 4, line 44: The imbalance between the # of teachers in the intervention (n=28) and control (n=4) groups could be a limitation. I suggest expanding the reason for this design choice and discussing its implications if this creates a limitation in the study.

Page 7, line 8: The imbalance between the number of teachers in the intervention (n=28) and control (n=4) groups is acknowledged, but its implications should be discussed more. The decision to limit quantitative analysis to within-group comparisons in the intervention arm, without exploratory contrasts with the control group, raises concerns about attributing observed changes to the program. I suggest framing the study as exploratory or preliminary, and discussing how this limitation affects internal validity and the strength of causal claims.

Page 7, line 22: The thematic analysis and triangulation are strong points of this study, but the description of the qualitative methods could be clearer. It would help to explain whether the coding was inductive, deductive, or a mix of both; how many researchers were involved in the analysis; how they handled disagreements during coding; and how the themes were connected to the quantitative results. If there is not enough space in the main text, I suggest adding this information in the supplementary material.

Page 7, line 37: Given that 94% of participants were women, it would be important to briefly reflect in the discussion on how this gender imbalance might have influenced the findings. Even if gender analysis was not part of the original design, acknowledging this limitation and its possible implications would strengthen the interpretation of the results.

Table 4: Table 4 is a valuable contribution to the manuscript. However, it would be clearer if the table explicitly stated what was observed in both the intervention and control groups for each domain (quantitative and qualitative columns). It is difficult to see exactly what is being integrated, since the reader doesn’t always know which group the qualitative insights refer to. I suggest including a mention of the control group findings and clarifying which results came from which group would make the integration summary more meaningful. Again, the imbalance in group sizes limits the strength of comparison, but acknowledging this within the table or its caption could help manage reader expectations

General: The manuscript suggests that the 3Cs programme offers a scalable model. However, this claim would need justification. Given the small sample size, the strong imbalance between intervention and control groups, and the absence of between-group comparisons, the study design does not yet provide robust evidence to support claims about scalability. I suggest either moderating this statement or adding more detail on what aspects of the intervention make it potentially scalable, even if the current findings are preliminary.

---

## [Reviewer Report]

I consider that it is a relevant paper that contributes to the use of “Contigo, Conmigo y con Todo -3C program”. The article is coherent and consistent because the objective is clearly stated. The introduction is updated and the authors make use of articles that have been written over the last 5 or 6 years. As for the method, the authors describe the sample and the selection criteria, explaining the instruments applied in the quantitative phase and they used focus groups for the qualitative phase with the Colombian teacher’s population. The procedure and the data analysis are also clearly stated. The authors used a mixed-method approach as the main results are depicted and according to the measures employed. The discussion explains the results obtained from the theoretical and investigative development as limitations of the study.

Although the quantitative results show changes related to increased resilience and decreased post-traumatic stress, the qualitative results show gains in professional identity purpose and compassionate practice, with the sample size of 32 teachers from eight schools, and the lack of homogenization, which lead us to question whether the results were due to the implementation of the program or other factors. Though a control group was proposed, it could not be taken into account in the analyses due to its size (n= 4), and despite the fact that measurements were taken at different times after the program was implemented during the follow-up period (6 and 9 months). The limitation of the study is related to the sample size which affects the generalization of the results as the authors indicated in the discussion. Furthermore, the sample size doesn´t allow for robust analyses that confirm real quantitative changes. These aspects also affect the validity and reliability of the results and it is a methodological constrain. Authors must justify aspects such as sample size and indicate how they can overcome this methodological limitation in the procedure.